# Censored Semi-Bandits: A Framework for Resource Allocation with Censored Feedback

**Arun Verma**
Department of IEOR
IIT Bombay, India
v.arun@iitb.ac.in

**Manjesh K. Hanawal**
Department of IEOR
IIT Bombay, India
mhanwal@iitb.ac.in

**Arun Rajkumar**
Department of CSE
IIT Madras, India
arunr@cse.iitm.ac.in

**Raman Sankaran**
LinkedIn India
Bengaluru, India
rsankara@linkedin.com

## Abstract

In this paper, we study *Censored Semi-Bandits*, a novel variant of the semi-bandits problem. The learner is assumed to have a fixed amount of resources, which it allocates to the arms at each time step. The loss observed from an arm is random and depends on the amount of resources allocated to it. More specifically, the loss equals zero if the allocation for the arm exceeds a constant (but unknown) threshold that can be dependent on the arm. Our goal is to learn a feasible allocation that minimizes the expected loss. The problem is challenging because the loss distribution and threshold value of each arm are unknown. We study this novel setting by establishing its 'equivalence' to Multiple-Play Multi-Armed Bandits (MP-MAB) and Combinatorial Semi-Bandits. Exploiting these equivalences, we derive optimal algorithms for our setting using the existing algorithms for MP-MAB and Combinatorial Semi-Bandits. Experiments on synthetically generated data validate performance guarantees of the proposed algorithms.

## 1 Introduction

Many real-life sequential resource allocation problems have a censored feedback structure. Consider, for instance, the problem of optimally allocating patrol officers (resources) across various locations in a city on a daily basis to combat *opportunistic* crimes. Here, a perpetrator picks a location (e.g., a deserted street) and decides to commit a crime (e.g., mugging) but does not go ahead with it if a patrol officer happens to be around in the vicinity. Though the true potential crime rate depends on the latent decision of the perpetrator, one observes feedback only when the crime is committed. Thus crimes that were planned but not committed get censored. This model of censoring is quite general and finds applications in several resource allocation problems such as police patrolling (Curtin et al., 2010), traffic regulations and enforcement (Adler et al., 2014; Rosenfeld and Kraus, 2017), poaching control (Nguyen et al., 2016; Gholami et al., 2018), supplier selection (Abernethy et al., 2016), advertisement budget allocation (Lattimore et al., 2014), among many others.

Existing approaches that deal with censored feedback in resource allocation problems fall into two broad categories. Curtin et al. (2010); Adler et al. (2014); Rosenfeld and Kraus (2017) learn good resource allocations from historical data. Nguyen et al. (2016); Gholami et al. (2018); Zhang et al. (2016); Sinha et al. (2018) pose the problem in a game-theoretic framework (opportunistic security games) and propose algorithms for optimal resource allocation strategies. While the first approach fails to capture the sequential nature of the problem, the second approach is agnostic to the user

(perpetrator) behavioral modeling. In this work, we balance these two approaches by proposing a simple yet novel threshold-based behavioral model, which we term as *Censored Semi Bandits* (CSB). The model captures how a user opportunistically reacts to an allocation.

In the first variation of our proposed behavioral model, we assume the threshold (user behavioral) is uniform across arms (locations). We establish that this setup is 'equivalent' to Multiple-Play Multi-Armed Bandits (MP-MAB), where a fixed number of arms is played in each round. We also study the more general variation where the threshold is arm dependent. We establish that this setup is equivalent to Combinatorial Semi-Bandits, where a subset of arms to be played is decided by solving a combinatorial 0-1 knapsack problem.

Formally, we tackle the sequential nature of the resource allocation problem by establishing its equivalence to the MP-MAB and Combinatorial Semi-Bandits framework. By exploiting this equivalence for our proposed threshold-based behavioral model, we develop novel resource allocation algorithms by adapting existing algorithms and provide optimal regret guarantees for the same.

**Related Work:** The problem of resource allocation for tackling crimes has received significant interest in recent times. Curtin et al. (2010) employ a static maximum coverage strategy for spatial police allocation while Nguyen et al. (2016) and Gholami et al. (2018) study game-theoretic and adversarial perpetrator strategies. We, on the other hand, restrict ourselves to a non-adversarial setting. (Adler et al., 2014) and Rosenfeld and Kraus (2017) look at traffic police resource deployment and consider the optimization aspects of the problem using real-time traffic, etc., which differs from the main focus of our work. Zhang et al. (2015) investigate dynamic resource allocation in the context of police patrolling and poaching for opportunistic criminals. Here they attempt to learn a model of criminals using a dynamic Bayesian network. Our approach proposes simpler and realistic modeling of perpetrators where the underlying structure can be exploited efficiently.

We pose our problem in the exploration-exploitation paradigm, which involves solving the MP-MAB and combinatorial 0-1 knapsack problem. It is different from the bandits with Knapsacks setting studied in Badanidiyuru et al. (2018), where resources get consumed in every round. The work of Abernethy et al. (2016) and Jain and Jamieson (2018) are similar to us in the sense that they are also threshold-based settings. However, the thresholding we employ naturally fits our problem and significantly differs from theirs. Specifically, their thresholding is on a sample generated from an underlying distribution, whereas we work in a Bernoulli setting where the thresholding is based on the allocation. Resource allocation with semi-bandits feedback (Lattimore et al., 2014, 2015; Dagan and Crammer, 2018) study a related but less general setup where the reward is based only on allocation and a hidden threshold. Our setting requires an additional unknown parameter for each arm, a 'mean loss,' which also affects the reward.

Allocation problems in the combinatorial setting have been explored in Cesa-Bianchi and Lugosi (2012); Chen et al. (2013); Rajkumar and Agarwal (2014); Combes et al. (2015); Chen et al. (2016); Wang and Chen (2018). Even though these are not related to our setting directly, we derive explicit connections to a sub-problem of our algorithm to the setup of Komiyama et al. (2015) and Wang and Chen (2018).

## 2   Problem Setting

We consider a sequential learning problem where $K$ denotes the number of arms (locations), and $Q$ denotes the amount of divisible resources. The loss at arm $i \in [K]$ where $[K] := \{1, 2, \ldots, K\}$, is Bernoulli distributed with rate $\mu_i \in (0, 1]$. Each arm can be assigned a fraction of resource that decides the feedback observed and the loss incurred from that arm – if the allocated resource is smaller than a certain threshold[1], the loss incurred is the realization of the arm, and it is observed. Otherwise, the realization is unobserved, and the incurred loss is zero. Let $\boldsymbol{a} := \{a_i : i \in [K]\}$, where $a_i \in [0, 1]$ denotes the resource allocated to arm $i$. For each $i \in [K]$, let $\theta_i \in (0, 1]$ denote the threshold associated with arm $i$ and is such that a loss is incurred at arm $i$ only if $a_i < \theta_i$. An allocation vector $\boldsymbol{a}$ is said to be feasible if $\sum_{i \in [K]} a_i \leq Q$ and set of all feasible allocations is denoted as $\mathcal{A}_Q$. The goal is to find a feasible resource allocation that results in a maximum reduction in the mean loss incurred.

In our setup, resources may be allocated to multiple arms. However, loss from each of the allocated arms may not be observed depending on the amount of resources allocated to them. We thus have a version of the partial monitoring system (Cesa-Bianchi et al., 2006; Bartók and Szepesvári, 2012; Bartók et al., 2014) with semi-bandit feedback, and we refer to it as censored semi-bandits (CSB). The vectors $\boldsymbol{\theta} := \{\theta_i\}_{i \in [K]}$ and $\boldsymbol{\mu} := \{\mu_j\}_{i \in [K]}$ are unknown and identify an instance of the CSB problem. Henceforth we identify a CSB instance as $P = (\boldsymbol{\mu}, \boldsymbol{\theta}, Q) \in [0,1]^K \times (0,1]^K \times \mathbb{R}_+$ and denote the collection of CSB instances as $\mathcal{P}_{\text{CSB}}$. As the number of arms $K = |\boldsymbol{\mu}|$, $K$ is known (implicitly) from an instance of CSB. For simplicity of discussion, we assume that $\mu_1 \leq \mu_2 \leq \ldots \leq \mu_K$, but the algorithms are not aware of this ordering. For instance $P \in \mathcal{P}_{\text{CSB}}$, the optimal allocation can be computed by the following 0-1 knapsack problem

$$\boldsymbol{a}^\star \in \arg\min_{\boldsymbol{a} \in \mathcal{A}_Q} \sum_{i=1}^K \mu_i \mathbb{1}_{\{a_i < \theta_i\}}. \tag{1}$$

Interaction between the environment and a learner is given in Algorithm 1.

---

**Algorithm 1** CSB Learning Protocol on instance $(\boldsymbol{\mu}, \boldsymbol{\theta}, Q)$

---

For each round $t$:

1. **Environment** generates a vector $\boldsymbol{X_t} = (X_{t,1}, X_{t,2}, \ldots, X_{t,K}) \in \{0,1\}^K$, where $\mathbb{E}[X_{t,i}] = \mu_i$ and the sequence $(X_{t,i})_{t \geq 1}$ is i.i.d. for all $i \in [K]$.

2. **Learner** picks an allocation vector $\boldsymbol{a}_t \in \mathcal{A}_Q$.

3. **Feedback and Loss:** The learner observes a random feedback $\boldsymbol{Y_t} = \{Y_{t,i} : i \in [K]\}$, where $Y_{t,i} = X_{t,i} \mathbb{1}_{\{a_{t,i} < \theta_i\}}$ and incurs loss $\sum_{i \in [K]} Y_{t,i}$.

---

The goal of the learner is to find a feasible resource allocation strategy at every round such that the cumulative loss is minimized. Specifically, we measure the performance of a policy that selects allocations $\{\boldsymbol{a}_t\}_{t \geq 1}$ in $T$ steps in terms of expected (pseudo) regret given by

$$\mathbb{E}[\mathcal{R}_T] = \mathbb{E}\left[\sum_{t=1}^T \sum_{i=1}^K X_{t,i} \mathbb{1}_{\{a_{t,i} < \theta_i\}}\right] - T \sum_{i=1}^K \mu_i \mathbb{1}_{\{a_i^\star < \theta_i\}}. \tag{2}$$

A good policy should have sub-linear expected regret, i.e., $\mathbb{E}[\mathcal{R}_T]/T \to 0$ as $T \to \infty$.

## 3 Identical Threshold for All Arms

In this section, we focus on the particular case of the CSB problem where $\theta_i = \theta_c$ for all $i \in [K]$. With abuse of notation, we continue to denote an instance of CSB with the same threshold as $(\boldsymbol{\mu}, \theta_c, Q)$, where $\theta_c \in (0,1]$ is the same threshold.

**Definition 1.** *For a given loss vector $\boldsymbol{\mu}$ and amount of resources $Q$, we say that thresholds $\theta_c$ and $\hat{\theta}_c$ are* allocation equivalent *if the following holds:*

$$\min_{\boldsymbol{a} \in \mathcal{A}_Q} \sum_{i=1}^K \mu_i \mathbb{1}_{\{a_i < \theta_c\}} = \min_{\boldsymbol{a} \in \mathcal{A}_Q} \sum_{i=1}^K \mu_i \mathbb{1}_{\{a_i < \hat{\theta}_c\}}.$$

Though $\theta_c$ can take any value in the interval $(0,1]$, an allocation equivalent to it can be confined to a finite set. The following lemma shows that the search for an allocation equivalent can be restricted to $\lceil K - Q + 1 \rceil$ elements.

**Lemma 1.** *For any $\theta_c \in (0,1]$ and $Q \geq \theta_c$, let $M = \min\{\lfloor Q/\theta_c \rfloor, K\}$ and $\hat{\theta}_c = Q/M$. Then $\theta_c$ and $\hat{\theta}_c$ are allocation equivalent. Further, $\hat{\theta}_c \in \Theta$ where $\Theta = \{Q/K, Q/(K-1), \cdots, \min\{1, Q\}\}$.*

Let $M = \min\{\lfloor Q/\theta_c \rfloor, K\}$. In the following, when arms are sorted in the increasing order of mean losses, we refer to the last $M$ arms as the *bottom-$M$* arms and the remaining arms as *top-$(K-M)$*

arms. It is easy to note that an optimal allocation with the same threshold $\theta_c$ is to allocate $\theta_c$ amount of resource to each of the bottom-$M$ arms and allocate the remaining resources to the other arms. Lemma 1 shows that range of allocation equivalent $\hat{\theta}_c$ for any instance $(\boldsymbol{\mu}, \theta_c, Q)$ is finite. Once this value is found, the problem reduces to identifying the bottom-$M$ arms and assigning resource $\hat{\theta}_c$ to each one of them to minimize the mean loss. The latter part is equivalent to solving a Multiple-Play Multi-Armed Bandits problem, as discussed next.

## 3.1 Equivalence to Multiple-play Multi-armed Bandits

In stochastic Multiple-Play Multi-Armed Bandits (MP-MAB), a learner can play a subset of arms in each round known as superarm (Anantharam et al., 1987). The size of each superarm is fixed (and known). The mean loss of a superarm is the sum of the means of its constituent arms. In each round, the learner plays a superarm and observes the loss from each arm played (semi-bandit feedback). The goal of the learner is to select a superarm that has the smallest mean loss. A policy in MP-MAB selects a superarm in each round based on the past information. Its performance is measured in terms of regret defined as the difference between cumulative loss incurred by the policy and that incurred by playing an optimal superarm in each round. Let $(\boldsymbol{\mu}, m) \in [0, 1]^K \times \mathbb{N}_+$ denote an instance of MP-MAB where $\boldsymbol{\mu}$ denotes the mean loss vector, and $m \leq K$ denotes the size of each superarm. Let $\mathcal{P}_{\text{CSB}}^c \subset \mathcal{P}_{\text{CSB}}$ denote the set of CSB instances with the same threshold for all arms. For any $(\boldsymbol{\mu}, \theta_c, Q) \in \mathcal{P}_{\text{CSB}}^c$ with $K$ arms and known threshold $\theta_c$, let $(\boldsymbol{\mu}, m)$ be an instance of MP-MAB with $K$ arms and each arm has the same Bernoulli distribution as the corresponding arm in the CSB instance with $m = K - M$, where $M = \min\{\lfloor Q/\theta_c \rfloor, K\}$ as earlier. Let $\mathcal{P}_{\text{MP}}$ denote the set of resulting MP-MAB problems and $f : \mathcal{P}_{\text{CSB}} \to \mathcal{P}_{\text{MP}}$ denote the above transformation.

Let $\pi$ be a policy on $\mathcal{P}_{\text{MP}}$. $\pi$ can also be applied on any $(\boldsymbol{\mu}, \theta_c, Q) \in \mathcal{P}_{\text{CSB}}^c$ with known $\theta_c$ to decide which set of arms are allocated resource as follows: in round $t$, let the information $(L_1, Y_1, L_2, Y_2, \ldots, L_{t-1}, Y_{t-1})$ collected on a CSB instance, where $L_s$ is the set of $K - M$ arms where no resource is applied and $Y_s$ is the samples observed from these arms. In round $t$, this information is given to $\pi$ which returns a set $L_t$ with $K - M$ elements. Then all arms other than arms in $L_t$ are given resource $\theta_c$. Let this policy on $(\boldsymbol{\mu}, \theta_c, Q) \in \mathcal{P}_{\text{CSB}}^c$ be denoted as $\pi'$. In a similar way a policy $\beta$ on $\mathcal{P}_{\text{CSB}}$ can be adapted to yield a policy for $\mathcal{P}_{\text{MP}}$ as follows: in round $t$, let the information $(S_1, Y_1, S_2, Y_2, \ldots, S_{t-1}, Y_{t-1})$ collected on an MP-MAB instance, where $S_s$ is the superarm played in round $s$ and $Y_s$ is the associated loss observed from each arms in $S_s$, is given to $\pi$ which returns a set $S_t$ of $K - M$ arms where no resources has to be applied. The superarm corresponding to $S_t$ is then played. Let this policy on $\mathcal{P}_{\text{MP}}$ be denoted as $\beta'$. Note that when $\theta_c$ is known, the mapping is invertible. The next proposition gives regret equivalence between the MP-MAB problem and CSB problem with a known same threshold.

**Proposition 1.** *Let $P := (\boldsymbol{\mu}, \theta_c, Q) \in \mathcal{P}_{CSB}^c$ with known $\theta_c$ then the regret of $\pi$ on $P$ is same as the regret of $\pi'$ on $f(P)$. Similarly, let $P' := (\boldsymbol{\mu}, m) \in \mathcal{P}_{MP}$, then the regret of a policy $\beta$ on $P'$ is same as the regret of $\beta'$ on $f^{-1}(P')$. Thus the set $\mathcal{P}_{CSB}$ with a known $\theta_c$ is 'regret equivalent' to $\mathcal{P}_{MP}$, i.e., $\mathcal{R}(\mathcal{P}_{CSB}^c) = \mathcal{R}(\mathcal{P}_{MP})$.*

**Lower bound**: As a consequence of the above equivalence and one-to-one correspondence between the MP-MAB and CSB with the same known threshold, a lower bound on MP-MAB is also a lower bound on CSB with the same threshold. Hence the following lower bound given for any strongly consistent algorithm (Anantharam et al., 1987, Theorem 3.1) is also a lower bound on the CSB problem with the same threshold:

$$\lim_{T \to \infty} \frac{\mathbb{E}[\mathcal{R}_T]}{\log T} \geq \sum_{i \in [K] \setminus [K-M]} \frac{\mu_i - \mu_{K-M}}{d(\mu_{K-M}, \mu_i)}, \tag{3}$$

where $d(p, q)$ is the Kullback-Leibler (KL) divergence between two Bernoulli distributions with parameter $p$ and $q$. Also note that we are in loss setting.

The above proposition suggests that any algorithm which works well for the MP-MAB also works well for the CSB once the threshold is known. Hence one can use algorithms like MP-TS (Komiyama et al., 2015) and ESCB (Combes et al., 2015) once an allocation equivalent to $\theta_c$ is found. MP-TS uses Thompson Sampling, whereas ESCB uses UCB (Upper Confidence Bound) and KL-UCB type indices. One can use any of these algorithms. But we adapt MP-TS to our setting as it gives the better empirical performance and is shown to achieve optimal regret bound for Bernoulli distributions.

## 3.2 Algorithm: CSB-ST

We develop an algorithm named **CSB-ST** for solving the Censored Semi Bandits problem with Same Threshold. It exploits the result in Lemma 1 and equivalence established in Proposition 1 to learn a good estimate of threshold and minimize the regret using a MP-MAB algorithm. **CSB-ST** consists of two phases, namely, threshold estimation and regret minimization.

---

**CSB-ST** Algorithm for solving the Censored Semi Bandits problem with Same Threshold

---

**Input:** $K, Q, \delta, \epsilon$

\\ **Threshold Estimation Phase** \\

1: Initialize $C = 0, l = 0, u = |\Theta|, i = \lceil u/2 \rceil$
2: Set $\Theta$ as Lemma 1, $T_{\theta_s} = 0, W_\delta = \log(\log_2(|\Theta|)/\delta)/(\max\{1, \lfloor Q \rfloor\} \log(1/(1 - \epsilon)))$
3: **while** $i \neq u$ **do**
4:      Set $\hat{\theta}_c = \Theta[i]$
5:      $A_t \leftarrow$ first $Q/\hat{\theta}_c$ arms. Allocate $\hat{\theta}_c$ resource to each arm $i \in A_t$
6:      If loss observed for any arm $i \in A_t$ then set $l = i, i = l + \lceil \frac{u-l}{2} \rceil, C = 0$ else $C = C + 1$
7:      If $C = W_\delta$ then set $u = i, i = u - \lfloor \frac{u-l}{2} \rfloor, C = 0$
8:      $T_{\theta_s} = T_{\theta_s} + 1$
9: **end while**

\\ **Regret Minimization Phase** \\

10: Set $M = Q/\hat{\theta}_c$ and $\forall i \in [K] : S_i = 1, F_i = 1$
11: **for** $t = T_{\theta_s} + 1, T_{\theta_s} + 2, \ldots, T$ **do**
12:      $\forall i \in [K] : \hat{\mu}_i(t) = \beta(S_i, F_i)$
13:      $L_t \leftarrow$ (K-M) arms with smallest estimates
14:      $\forall i \in L_t :$ allocate no resource to arm $i$. $\forall j \in K \setminus L_t :$ allocate $\hat{\theta}_c$ resources to arm $j$
15:      $\forall i \in L_t :$ observe $X_{t,i}$. Update $S_i = S_i + X_{t,i}$ and $F_i = F_i + 1 - X_{t,i}$
16: **end for**

---

**Threshold Estimation Phase:** This phase finds a threshold $\hat{\theta}_c$ that is allocation equivalent to the underlying threshold $\theta_c$ with high probability by doing a binary search over the set $\Theta = \{Q/K, Q/(K-1), \ldots, \min\{1, Q\}\}$. The elements of $\Theta$ are arranged in increasing order and are candidates for $\theta_c$. The search starts by taking $\hat{\theta}_c$ to be the middle element in $\Theta$ and allocating $\hat{\theta}_c$ resource to the first $Q/\hat{\theta}_c$ arms (denoted as $A_t$ in Line 5). If a loss is observed at any of these arms, it implies that $\hat{\theta}_c$ is an underestimate of $\theta_c$. All the candidates lower than the current value of $\hat{\theta}_c$ in $\Theta$ are eliminated, and the search is repeated in the remaining half of the elements again by starting with the middle element (Line 6). If no loss is observed for consecutive $W_\delta$ rounds, then $\hat{\theta}_i$ is possibly an overestimate. Accordingly, all the candidates larger than the current value of $\hat{\theta}_c$ in $\Theta$ are eliminated, and the search is repeated starting with the middle element in the remaining half (Line 7). The variable $C$ keeps track of the number of the consecutive rounds for which no loss is observed. It changes to 0 either after observing a loss or if no loss is observed for consecutive $W_\delta$ rounds.

Note that if $\hat{\theta}_c$ is an underestimate and no loss is observed for consecutive $W_\delta$ rounds, then $\hat{\theta}_c$ will be reduced, which leads to a wrong estimate of $\hat{\theta}_c$. To avoid this, we set the value of $W_\delta$ such that the probability of happening of such an event is upper bounded by $\delta$. The next lemma gives a bound on the number of rounds needed to find allocation equivalent for threshold $\theta_c$ with high probability.

**Lemma 2.** *Let $(\boldsymbol{\mu}, \theta_c, Q)$ be an CSB instance such that $\mu_1 \geq \epsilon > 0$. Then with probability at least $1 - \delta$, the number of rounds needed by the threshold estimation phase of* **CSB-ST** *to find the allocation equivalent for threshold $\theta_c$ is bounded as*

$$T_{\theta_s} \leq \frac{\log(\log_2(|\Theta|)/\delta)}{\max\{1, \lfloor Q \rfloor\} \log(1/(1 - \epsilon))} \log_2(|\Theta|).$$

Once $\hat{\theta}_c$ is known, $\boldsymbol{\mu}$ needs to be estimated. The resources can be allocated such that no losses are observed for maximum $M$ arms. As our goal is to minimize the mean loss, we have to select $M$ arms with highest mean loss and then allocate $\hat{\theta}_c$ to each of them. It is equivalent to find $K - M$ arms with

the least mean loss then allocate no resources to these arms and observe their losses. These losses are then used for updating the empirical estimate of the mean loss of arms.

**Regret Minimization Phase:** The regret minimization phase of **CSB-ST** adapts Multiple-Play Thompson Sampling (MP-TS) (Komiyama et al., 2015) for our setting. It works as follows: initially we set the prior distribution of each arm as the Beta distribution $\beta(1,1)$, which is same as Uniform distribution on $[0,1]$. $S_i$ represents the number of rounds when loss is observed whereas $F_i$ represents the number of round when loss is not observed. Let $S_i(t)$ and $F_i(t)$ denote the values of $S_i$ and $F_i$ in the beginning of round $t$. In round $t$, a sample $\hat{\mu}_i$ is independently drawn from $\beta(S_i(t), F_i(t))$ for each arm $i \in [K]$. $\hat{\mu}_i$ values are ranked by their increasing values. The first $K-M$ arm are assigned no resources (denoted as set $L_t$ in Line 13) while each of the remaining $M$ arms are allocated $\hat{\theta}_c$ resources. The loss $X_{t,i}$ is observed for each arm $i \in L_t$ and then success and failure counts are updated by setting $S_i = S_i + X_{t,i}$ and $F_i = F_i + 1 - X_{t,i}$.

### 3.2.1 Regret Upper Bound

For instance $(\boldsymbol{\mu}, \theta, Q)$ and any feasible allocation $\boldsymbol{a} \in \mathcal{A}_Q$, we define $\nabla_{\boldsymbol{a}} = \sum_{i=1}^K \mu_i \big( \mathbb{1}_{\{a_i < \theta_i\}} - \mathbb{1}_{\{a_i^\star < \theta_i\}} \big)$ and $\nabla_m = \max_{\boldsymbol{a} \in \mathcal{A}_Q} \nabla_{\boldsymbol{a}}$. We are now ready to state the regret bounds.

**Theorem 1.** *Let* $\mu_1 \geq \epsilon > 0$, $W_\delta = \log(\log_2(|\Theta|)/\delta)/\max\{1, \lfloor Q \rfloor\} \log(1/(1-\epsilon))$, $\mu_{K-M} < \mu_{K-M+1}$, *and* $T > W_\delta \log_2(|\Theta|)$. *Set* $\delta = T^{-(\log T)^{-\alpha}}$ *in* **CSB-ST** *such that* $\alpha > 0$. *Then the regret of* **CSB-ST** *is upper bounded as*

$$\mathbb{E}\left[\mathcal{R}_T\right] \leq W_\delta \log_2\left(|\Theta|\right) \nabla_m + O\left((\log T)^{2/3}\right) + \sum_{i \in [K] \setminus [K-M]} \frac{(\mu_i - \mu_{K-M}) \log T}{d(\mu_{K-M}, \mu_i)}.$$

The first term in the regret bound of Theorem 1 corresponds to the length of the threshold estimation phase, and the remaining parts correspond to the expected regret in the regret minimization phase.

Note that the assumption $\mu_1 \geq \epsilon$ is only required to guarantee that the threshold estimation phase terminates in finite number of rounds. This assumption is not needed to get the bound on expected regret in the regret minimization phase. The assumption $\mu_{K-M} < \mu_{K-M+1}$ ensures that Kullback-Leibler divergence in the bound is well defined. This assumption is also equivalent to assuming that the set of top-$M$ arms is unique.

**Corollary 1.** *The regret of* **CSB-ST** *is asymptotically optimal.*

Note that $W_\delta = O\left((\log T)^{1-\alpha}\right)$ for any $\alpha > 0$ and $\delta = T^{-(\log T)^{-\alpha}}$ in **CSB-ST**. Now the proof of Corollary 1 follows by comparing the above bound with the lower bound given in Eq. (3).

## 4  Different Thresholds

In this section, we consider a more difficult problem where the threshold may not be the same for all arms. Let $KP(\boldsymbol{\mu}, \boldsymbol{\theta}, Q)$ denote a 0-1 knapsack problem with capacity $Q$ and $K$ items where item $i$ has weight $\theta_i$ and value $\mu_i$. Our next result gives the optimal allocation for an instance in $\mathcal{P}_{\text{CSB}}$.

**Proposition 2.** *Let* $P = (\boldsymbol{\mu}, \boldsymbol{\theta}, Q) \in \mathcal{P}_{\text{CSB}}$. *Then the optimal allocation for* $P$ *is a solution of* $KP(\boldsymbol{\mu}, \boldsymbol{\theta}, Q)$ *problem.*

The proof of the above proposition and computational issues of the 0-1 knapsack with fractional values are given in the supplementary. We next discuss the condition when two threshold vectors are allocation equivalent. Extending the definition of allocation equivalence to threshold vectors, we say that two vectors $\boldsymbol{\theta}$ and $\hat{\boldsymbol{\theta}}$ are allocation equivalent if minimum mean loss in instances $(\boldsymbol{\mu}, \boldsymbol{\theta}, Q)$ and $(\boldsymbol{\mu}, \hat{\boldsymbol{\theta}}, Q)$ are the same for any loss vector $\boldsymbol{\mu}$ and resource $Q$. This equivalence allows us to look for estimated thresholds within some tolerance. We need the following notations to formalize this notion.

For an instance $P := (\boldsymbol{\mu}, \boldsymbol{\theta}, Q)$, recall that $\boldsymbol{a}^\star = (a_1^\star, \ldots, a_K^\star)$ denotes the optimal allocation. Let $r = Q - \sum_{i: a_i^\star \geq \theta_i} \theta_i$, where $r$ is the residual resources after the optimal allocation. Define $\gamma := r/K$. Any problem instance with $\gamma = 0$ becomes a 'hopeless' problem instance as the only vector that is allocation equivalent to $\boldsymbol{\theta}$ is $\boldsymbol{\theta}$ itself, which demands $\theta_i$ values to be estimated with full precision to achieve optimal allocation. However, if $\gamma > 0$, an optimal allocation can still be found with small errors in the estimates of $\theta_i$ as shown next.

**Lemma 3.** *Let $\gamma > 0$ and $\forall i \in [K] : \hat{\theta}_i \in [\theta_i, \theta_i + \gamma]$. Then for any $\boldsymbol{\mu} \in [0,1]^K$ and $Q$, the $\boldsymbol{\theta}$ and $\hat{\boldsymbol{\theta}}$ are allocation equivalent.*

The proof follows by an application of Theorem 3.2 in Hifi and Mhalla (2013) which gives conditions for two weight vectors $\boldsymbol{\theta}_1$ and $\boldsymbol{\theta}_2$ to have the same solution in $KP(\boldsymbol{\mu}, \boldsymbol{\theta}_1, Q)$ and $KP(\boldsymbol{\mu}, \boldsymbol{\theta}_2, Q)$ for any $\boldsymbol{\mu}$ and $Q$. Once we accurately estimate the threshold $\boldsymbol{\theta}$ so that the estimate $\hat{\boldsymbol{\theta}}$ is an allocation equivalent of $\boldsymbol{\theta}$, the problem is equivalent to solving the $KP(\boldsymbol{\mu}, \hat{\boldsymbol{\theta}}, Q)$. The latter part is equivalent to solving a Combinatorial Semi-Bandits as we establish next. Combinatorial Semi-Bandits are the generalization of MP-MAB, where the size of the superarms need not be the same in each round.

**Proposition 3.** *The CSB problem with known threshold vector $\boldsymbol{\theta}$ is regret equivalent to a Combinatorial Semi-Bandits where Oracle uses $KP(\boldsymbol{\mu}, \boldsymbol{\theta}, Q)$ to identify the optimal subset of arms.*

### 4.1   Algorithm: CSB-DT

We develop an algorithm named **CSB-DT** for solving the Censored Semi Bandits problem with Different Threshold. It exploits the result of Lemma 3 and equivalence established in Proposition 3 to learn a good estimate of the threshold for each arm and minimizes the regret using an algorithm from Combinatorial Semi-Bandits. **CSB-DT** also consists of two phases: threshold estimation and regret minimization.

---

**CSB-DT** Algorithm for solving the Censored Semi Bandits problem with Different Threshold

---

**Input:** $K, Q, \delta, \epsilon, \gamma$

\\ **Threshold Estimation Phase** \\

1: Initialize $\forall i \in [K] : \theta_{l,i} = 0, \theta_{u,i} = 1, \theta_{g,i} = 0, C_i = 0$.
2: Set $T_{\theta_d} = 0, W_\delta = \log(K \log_2(\lceil 1 + 1/\gamma \rceil)/\delta)/\log(1/(1-\epsilon))$
3: $\forall i \in [\lfloor Q \rfloor]$ : allocate $\hat{\theta}_i = 0.5$ resource. $\forall j \in [\lfloor Q \rfloor + 1, K]$ : allocate $\hat{\theta}_j = \frac{Q - \lfloor Q \rfloor/2}{K - \lfloor Q \rfloor}$ resources
4: **while** $\theta_{g,i} = 0$ for any $i \in [K]$ **do**
5:     **for** $i = 1, \ldots, K$ **do**
6:         **if** loss observe for arm $i$ with $\theta_{g,i} = 0$ and $\theta_{l,i} < \hat{\theta}_i$ **then**
7:             Set $\theta_{l,i} = \hat{\theta}_i, \hat{\theta}_i = (\theta_{u,i} + \theta_{l,i})/2, C_i = 0$. If available allocate resource $\hat{\theta}_i$
8:         **else**
9:             If allocated resources is $\hat{\theta}_i$ then reset $C_i = C_i + 1$
10:             **if** $C_i = W_\delta$ and $\theta_{g,i} = 0$ **then**
11:                 Set $\theta_{u,i} = \hat{\theta}_i, \hat{\theta}_i = (\theta_{u,i} + \theta_{l,i})/2, C_i = 0$. If available allocate resource $\hat{\theta}_i$
12:                 If $\theta_{u,i} - \theta_{l,i} \leq \gamma$ then set $\theta_{g,i} = 1$ and $\hat{\theta}_i = \theta_{u,i}$
13:             **end if**
14:         **end if**
15:     **end for**
16:     **while** free resources are available **do**
17:         Allocate $\hat{\theta}_i$ resources to a new randomly chosen arm $i$ from the arms having $\theta_{g,i} = 1$
18:     **end while**
19:     $T_{\theta_d} = T_{\theta_d} + 1$
20: **end while**

\\ **Regret Minimization Phase** \\

21: $\forall i \in [K] : S_i = 1, F_i = 1$
22: **for** $t = T_{\theta_d} + 1, T_{\theta_d} + 2, \ldots, T$ **do**
23:     $\forall i \in [K] : \hat{\mu}_i(t) \leftarrow \text{Beta}(S_i, F_i)$
24:     $L_t \leftarrow \text{Oracle}\big(KP(\hat{\boldsymbol{\mu}}(t), \hat{\boldsymbol{\theta}}, Q)\big)$
25:     $\forall i \in L_t$ : allocate no resource to arm $i$. $\forall j \in K \setminus L_t$ : allocate $\hat{\theta}_j$ resources to arm $j$
26:     $\forall i \in L_t$ : observe $X_{t,i}$. Update $S_i = S_i + X_{t,i}$ and $F_i = F_i + 1 - X_{t,i}$
27: **end for**

---

**Threshold Estimation Phase:** This phase finds a threshold that is allocation equivalent of $\boldsymbol{\theta}$ with high probability. This is achieved by ensuring that $\hat{\theta}_i \in [\theta_i, \theta_i + \gamma]$ for all $i$ (Lemma 3). For each arm

$i \in [K]$ a binary search is performed over the interval $[0, 1]$ by maintaining variables $\hat{\theta}_i, \theta_{l,i}, \theta_{u,i}, \theta_{g,i}$, and $C_i$ where $\hat{\theta}_i$ is the current estimate of $\theta_i$; $\theta_{l,i}$ and $\theta_{u,i}$ denote the lower and upper bound of the binary search region for arm $i$; and $\theta_{g,i}$ indicates whether current estimate lies in the interval $[\theta_i, \theta_i + \gamma]$. In each round, the threshold estimate of arms are first updated sequentially and then tested on their respective arms. $C_i$ keeps count of consecutive rounds without no loss for $\hat{\theta}_i$. It changes to 0 either after observing a loss or if no loss is observed for consecutive $W_\delta$ rounds.

The threshold estimation phase starts with allocating $0.5$ resource for first $\lfloor Q \rfloor$ arms and $(Q - \lfloor Q \rfloor/2)/(K - \lfloor Q \rfloor)$ for the remaining arms (Line 3). In each round, allocated resource are applied on each arm and based on the observations, their estimates and the allocated resource are updated sequentially starting from 1 to $K$ as follows. If a loss is observed for arm $i$ having bad threshold estimate ($\theta_{g,i} = 0$) and $\theta_{l,i} < \hat{\theta}_i$, then it implies that $\hat{\theta}_i$ is an underestimate of $\theta_i$ and the following actions are performed – 1) lower end of search region is increased to $\hat{\theta}_i$, i.e., $\theta_{l,i} = \hat{\theta}_i$; 2) its estimate $\hat{\theta}_i$ is set to $(\theta_{u,i} + \theta_{l,i})/2$; 3) if available allocate $\hat{\theta}_i$ resource to arm $i$; and 4) set $C_i = 0$ (Line 7).

If no loss is observed after allocating $\hat{\theta}_i$ resources for $W_\delta$ successive rounds for arm $i$ with bad threshold estimate, then it implies that $\hat{\theta}_i$ is overestimated and following actions are performed – 1) the upper end of the search region is changed to $\hat{\theta}_i$, i.e, $\theta_{u,i} = \hat{\theta}_i$; 2) its estimate $\hat{\theta}_i$ is set to $(\theta_{u,i} + \theta_{l,i})/2$; and 3) if available allocate $\hat{\theta}_i$ resource to arm $i$ (Line 11). Further, whether goodness of $\hat{\theta}_i$ holds, i.e., $\hat{\theta}_i \in [\theta_i, \theta_i + \gamma]$ is checked by condition $\theta_{u,i} - \theta_{l,i} \le \gamma$. If the condition holds, the threshold estimation of arm is within desired accuracy and this is indicated by setting $\theta_{g,i}$ to 1 and $\hat{\theta}_i = \theta_{u,i}$ (Line 12). Any unassigned resources are given to randomly chosen arms having good threshold estimates (all arms with $\theta_{g,i} = 1$) where each arm $i$ gets only $\hat{\theta}_i$ resources (Line 17).

The value of $W_\delta$ in **CSB-DT** is set such that the probability of estimated threshold does not lie in $[\theta_i, \theta_i + \gamma]$ for all arms is upper bounded by $\delta$. Following lemma gives the bounds on the number of rounds needed to find the allocation equivalent for threshold vector $\boldsymbol{\theta}$ with high probability.

**Lemma 4.** *Let $(\boldsymbol{\mu}, \boldsymbol{\theta}, Q)$ be an instance of CSB such that $\gamma > 0$ and $\mu_1 \ge \epsilon > 0$. Then with probability at least $1 - \delta$, the number of rounds needed by threshold estimation phase of* **CSB-DT** *to find the allocation equivalent for threshold vector $\boldsymbol{\theta}$ is bounded as*

$$T_{\theta_d} \le \frac{K \log(K \log_2(\lceil 1 + 1/\gamma \rceil)/\delta)}{\max\{1, \lfloor Q \rfloor\} \log(1/(1-\epsilon))} \log_2(\lceil 1 + 1/\gamma \rceil).$$

**Regret Minimization Phase:** For this phase, we could use an algorithm that works well for the Combinatorial Semi-Bandits, like SDCB (Chen et al., 2016) and CTS (Wang and Chen, 2018). CTS uses Thompson Sampling, whereas SDCB uses the UCB type index. We adapt the CTS to our setting due to better empirical performance. This phase is similar to the regret minimization phase of CSB-ST except that superarm to play is selected by Oracle that uses $KL(\hat{\boldsymbol{\mu}}(t), \hat{\boldsymbol{\theta}}, Q)$ to identify the arms where the learner has to allocate no resources.

### 4.1.1 Regret Upper Bound

Let $\nabla_{\boldsymbol{a}}$ and $\nabla_m$ be defined as in Section 3.2.1. Let $\gamma > 0$, $S_{\boldsymbol{a}} = \{i : a_i < \theta_i\}$ for any feasible allocation $\boldsymbol{a}$ and $k^\star = |S_{\boldsymbol{a}^\star}|$. We redefine $W_\delta = \log(K \log_2(\lceil 1 + 1/\gamma \rceil)/\delta)/\log(1/(1-\epsilon))$.

**Theorem 2.** *Let $(\boldsymbol{\mu}, \boldsymbol{\theta}, Q) \in \mathcal{P}_{CSB}$ such that $\gamma > 0$, $\mu_1 \ge \epsilon$, and $T > W_\delta \log_2(\lceil 1 + 1/\gamma \rceil)$. Set $\delta = T^{-(\log T)^{-\alpha}}$ in* **CSB-DT** *such that $\alpha > 0$. Then the expected regret of* **CSB-DT** *is upper bounded as*

$$\mathbb{E}\left[\mathcal{R}_T\right] \le \left(\frac{KW_\delta \log_2\left(\lceil 1 + 1/\gamma \rceil\right)}{\max\{1, \lfloor Q \rfloor\}}\right) \nabla_m + \left(\sum_{i \in [K]} \max_{S_{\boldsymbol{a}} : i \in S_{\boldsymbol{a}}} \frac{8|S_{\boldsymbol{a}}| \log T}{\nabla_{\boldsymbol{a}} - 2(k^{\star 2} + 2)\eta}\right) +$$

$$\left(\frac{K(K-Q)^2}{\eta^2} + \frac{8\alpha_1}{\eta^2} \left(\frac{4}{\eta^2} + 1\right)^{k^\star} \log \frac{k^\star}{\eta^2} + 3K\right) \nabla_m,$$

*for any $\eta$ such that $\forall \boldsymbol{a} \in \mathcal{A}_Q, \nabla_{\boldsymbol{a}} > 2(k^{\star 2} + 2)\eta$ and a problem independent constant $\alpha_1$.*

The first term of expected regret is due to the threshold estimation phase. Threshold estimation takes $T_{\theta_d}$ rounds to complete, and $\nabla_m$ is the maximum regret that can be incurred in any round. Then the

maximum regret due to threshold estimation is bounded by $T_{\theta_d} \nabla_m$. The remaining terms correspond to the regret due to the regret minimization phase. Further, the expected regret of **CSB-DT** can be shown be $\mathbb{E}\left[\mathcal{R}_T\right] \leq O(K \log T / \nabla_{\min})$, where $\nabla_{\min}$ is the minimum gap between the mean loss of optimal allocation and any non-optimal allocation.

## 5 Experiments

We ran computer simulations to evaluate the empirical performance of proposed algorithms. Our simulations involve two synthetically generated instances. In Instance 1, the threshold is the same for all arm, whereas in Instance 2, it varies across arms. The details of the instances are as follows:

**Instance 1 (Identical Threshold):** It has $K = 20, Q = 7, \theta_c = 0.7, \delta = 0.1, \epsilon = 0.1$ and $T = 10000$. The loss of arm $i$ is Bernoulli distribution with parameter $0.25 + (i-1)/50$.
**Instance 2 (Different Thresholds):** It has $K = 5, Q = 2, \delta = 0.1, \epsilon = 0.1, \gamma = 10^{-3}$ and $T = 5000$. The mean loss vector is $\boldsymbol{\mu} = [0.9, 0.89, 0.87, 0.58, 0.3]$ and corresponding threshold vector is $\boldsymbol{\theta} = [0.7, 0.7, 0.7, 0.6, 0.35]$. The loss of arm $i$ is Bernoulli distributed with parameter $\mu_i$.

For Instance 1, we only varied the number of resource $Q$ and observed the regret of **CSB-ST** as given in Fig. 1. We observe that when resources are small, the learner can allocate resources to a few arms but observes loss from more arms. On the other hand, when resources are more, the learner allocates resources to more arms and observes loss from fewer arms. Thus as resources increase, we move from semi-bandit feedback to bandit feedback and hence regret increases with increase in the amount of resources. Next, we only varied $\theta_c$ in Instance 1, and the regret of **CSB-ST** is shown in Fig. 2. Similar trends are observed as the decrease in threshold leads to an increase in the number of arms that can be allocated resources and vice-versa. Therefore the amount of feedback decreases as the threshold decreases and leads to more regret. We repeated the experiment 100 times and plotted the regret with a 95% confidence interval (the vertical line on each curve shows the confidence interval). The empirical results validate sub-linear bounds for our algorithms.

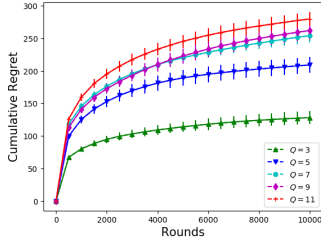
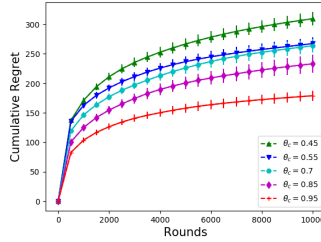
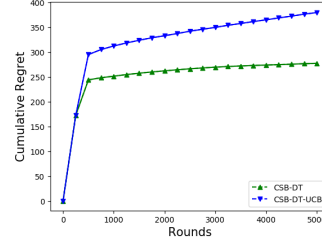

**Figure 1:** Varying amount of resources.　　**Figure 2:** Varying value of same threshold.　　**Figure 3:** UCB and TS based Algorithms.

We also compare the performance of **CSB-DT** against the CSB-DT-UCB algorithm, which uses the UCB type index as used in the SDCB algorithm (Chen et al., 2016) on Instance 2. As shown in Fig. 3, as expected, Thompson Sampling (TS) based **CSB-DT** outperforms its UCB based counterpart CSB-DT-UCB. The pseudo-code of CSB-DT-UCB is given in the supplementary material.

## 6 Conclusion and Future Extensions

In this work, we proposed a novel framework for resource allocation problems using a variant of semi-bandits named censored semi-bandits. In our setup, loss observed from an arm depends on the amount of resource allocated, and hence, the loss can be censored. We consider a threshold-based model where loss from an arm is observed when allocated resource is below a threshold. The goal is to assign a given resource to arms such that total expected loss is minimized. We considered two variants of the problem, depending on whether or not the thresholds are the same across the arms. For the variant where thresholds are the same across the arms, we established that it is equivalent to the Multiple-Play Multi-Armed Bandit problem. For the second variant where threshold can depend on the arm, we established that it is equivalent to a more general Combinatorial Semi-Bandit problem. Exploiting these equivalences, we developed algorithms that enjoy optimal performance guarantees.

We decoupled the problem of threshold and mean loss estimation. It would be interesting to explore if this can be done jointly, leading to better performance guarantees. Another new extension of work is to relax the assumptions that mean losses are strictly positive, and time horizon $T$ is known.

## Acknowledgments

Arun Verma would like to thank travel support from Google and NeurIPS. Manjesh K. Hanawal would like to thank the support from INSPIRE faculty fellowships from DST, Government of India, SEED grant (16IRCCSG010) from IIT Bombay, and Early Career Research (ECR) Award from SERB. Initial discussions of this work were done when Raman Sankaran was at Conduent Labs India.

## Footnotes

[1]One could consider a smooth function instead of a step function, but the analysis is more involved, and our results need not generalize straightforwardly.

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
