[Supplementary Material]

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

# Supplementary Material for 'Censored Semi-Bandits: A Framework for Resource Allocation with Censored Feedback'

## A  Proofs related to Section 3

### A.1  Proof of Lemma 1

**Lemma 1.** *For any $\theta_c \in (0,1]$ and $Q \geq \theta_c$, let $M = \min\{\lfloor Q/\theta_c \rfloor, K\}$ and $\hat{\theta}_c = Q/M$. Then $\theta_c$ and $\hat{\theta}_c$ are allocation equivalent. Further, $\hat{\theta}_c \in \Theta$ where $\Theta = \{Q/K, Q/(K-1), \cdots, \min\{1, Q\}\}$.*

*Proof.* The case $\lfloor Q/\theta_c \rfloor \geq K$ is trivial. We consider the case $\lfloor Q/\theta_c \rfloor < K$. By definition $M = \min\{\lfloor Q/\theta_c \rfloor, K\}$. We have $M \leq Q/\theta_c$ and $\theta_c \leq Q/M \doteq \hat{\theta}_c$. Hence $\hat{\theta}_c \geq \theta_c$. Therefore $\hat{\theta}_c$ fraction of resource allocation at a location has same reduction in the mean loss as $\theta_c$. Further, in both the instances $(\boldsymbol{\mu}, \theta_c, Q)$ and $(\boldsymbol{\mu}, \hat{\theta}_c, Q)$ the optimal allocations incur no loss from the bottom-$M$ and the same amount of loss from the top $K - M$ arms. Hence the mean loss reduction for both the instances is same. This completes the proof of first part. As $M \in \{1, \ldots, K\}$ and $\hat{\theta}_c \leq 1$, the possible value of $\hat{\theta}_c$ is only one of the elements in the set $\Theta = \{Q/K, Q/(K-1), \cdots, \min\{1, Q\}\}$.  $\square$

### A.2  Proof of Lemma 2

**Lemma 2.** *Let $(\boldsymbol{\mu}, \theta_c, Q)$ be an CSB instance such that $\mu_1 \geq \epsilon > 0$. Then with probability at least $1 - \delta$, the number of rounds needed by the threshold estimation phase of **CSB-ST** to find the allocation equivalent for threshold $\theta_c$ is bounded as*

$$T_{\theta_s} \leq \frac{\log(\log_2(|\Theta|)/\delta)}{\max\{1, \lfloor Q \rfloor\} \log\left(1/(1-\epsilon)\right)} \log_2(|\Theta|).$$

*Proof.* When $\hat{\theta}_c < \theta_c$, it is possible that no loss is observed for $W_\delta$ consecutive rounds that leads to incorrect estimation of $\theta_c$. We want to set $W_\delta$ in such a way that the probability of occurring of such event is upper bounded by $\delta$. This probability is bounded as follows:

$$\mathbb{P}\left\{\text{No loss is observed on } Q/\hat{\theta}_c \text{ arms for } W_\delta \text{ consecutive rounds at } \hat{\theta}_c < \theta_c \text{ (underestimate)}\right\}$$

$$= \prod_{i \leq Q/\hat{\theta}_i} (1 - \mu_i)^{W_\delta} \quad \text{(as } (1 - \mu_i) \text{ is the probability of not observing loss at location } i\text{)}$$

$$\leq \prod_{i \leq M} (1 - \mu_i)^{W_\delta} \quad \left(\text{as } \hat{\theta}_c < \theta_c \implies M \leq Q/\hat{\theta}_c\right)$$

$$\leq \prod_{i \leq M} (1 - \epsilon)^{W_\delta} = (1 - \epsilon)^{MW_\delta} \leq (1 - \epsilon)^{QW_\delta} \quad \left(\text{as } M \geq Q\right)$$

Since we are using binary search, the algorithm goes through at most $\log_2(|\Theta|)$ underestimates of $\theta_c$. Let $I$ denote the set of indices of these underestimates in $\Theta$

$$\mathbb{P}\left\{\text{No loss is observed for consecutive } W_\delta \text{ rounds at any underestimate of } \theta_c\right\}$$

$$\leq \sum_{i \in I} \mathbb{P}\left\{\text{No loss is observed for consecutive } W_\delta \text{ rounds at the underestimate } \Theta(i)\right\}$$

$$\leq (1 - \epsilon)^{QW_\delta} \log_2(|\Theta|)$$

As we are interesting in bounding the probability of making mistake by $\delta$, we get,

$$(1 - \epsilon)^{QW_\delta} \log_2(|\Theta|) \leq \delta$$

$$\implies (1-\epsilon)^{QW_\delta} \le \delta/\log_2(|\Theta|)$$

Taking log on both side of above equation, we get

$$QW_\delta \log(1-\epsilon) \le \log(\delta/\log_2(|\Theta|)) \implies QW_\delta \log\left(\frac{1}{1-\epsilon}\right) \ge \log(\log_2(|\Theta|)/\delta)$$

$$\implies W_\delta \ge \frac{\log(\log_2(|\Theta|)/\delta)}{Q\log\left(\frac{1}{1-\epsilon}\right)}$$

We set

$$W_\delta = \frac{\log(\log_2(|\Theta|)/\delta)}{Q\log\left(\frac{1}{1-\epsilon}\right)} \tag{4}$$

Hence, the minimum rounds needed to find $\hat{\theta}_c$ with probability at least $1-\delta$ is $W_\delta \log_2(|\Theta|)$. $\qquad\square$

### A.3 Proof of Proposition 1

**Proposition 1.** *Let $P := (\boldsymbol{\mu}, \theta_c, Q) \in \mathcal{P}^c_{CSB}$ with known $\theta_c$ then the regret of $\pi$ on $P$ is same as the regret of $\pi'$ on $f(P)$. Similarly, let $P' := (\boldsymbol{\mu}, m) \in \mathcal{P}_{MP}$, then the regret of a policy $\beta$ on $P'$ is same as the regret of $\beta'$ on $f^{-1}(P')$. Thus the set $\mathcal{P}_{CSB}$ with a known $\theta_c$ is 'regret equivalent' to $\mathcal{P}_{MP}$, i.e., $\mathcal{R}(\mathcal{P}^c_{CSB}) = \mathcal{R}(\mathcal{P}_{MP})$.*

*Proof.* Let $\pi$ be a policy on $P := (\boldsymbol{\mu}, \theta_c, Q) \in \mathcal{P}^c_{CSB}$. The regret of policy $\pi$ on $P$ is given by

$$\mathcal{R}_T(\pi, P) = \sum_{t=1}^{T} \left( \sum_{i=1}^{K} \mu_i \mathbb{1}_{\{a_{t,i} < \theta_c\}} - \sum_{i=1}^{K} \mu_i \mathbb{1}_{\{a_i^\star < \theta_c\}} \right),$$

where $\boldsymbol{a}^\star$ is the optimal allocation for $P$. Consider $f(P) = (\boldsymbol{\mu}, m) \in \mathcal{P}_{MP}$ where $\boldsymbol{\mu}$ is same as in $P$ and $m = K - M$, where $M = \min\{\lfloor Q/\theta_c \rfloor, K\}$. The regret of policy $\pi'$ on $f(P)$ is given by

$$\mathcal{R}_T(\pi', f(P)) = \sum_{t=1}^{T} \left( \sum_{i \in S_t} \mu_i - \sum_{i=1}^{K-M} \mu_i \right),$$

where $S_t$ is the superarm played in round $t$. Recall the ordering $\mu_1 \le \mu_2 \le \ldots \le \mu_K$. It is clear that $\sum_{i=1}^{K} \mu_i \mathbb{1}_{\{a_i^\star < \theta_c\}} = \sum_{i=1}^{K-M} \mu_i$. Let $L_t$ be the set of arms where no resources are allocated by $\pi$ in round $t$. Since, loss only incurred from arms in the set $L_t$, we have $\sum_{i=1}^{K} \mu_i \mathbb{1}_{\{a_{t,i} < \theta_c\}} = \sum_{i \in L_t} \mu_i$. From the definition of $\pi'$, notice that in round $t$, $\pi'$ selects superarm $S_t = L_t$, i.e., set of arms returned by $\pi$ for which no resourced are applied. Hence $\sum_{i=1}^{K} \mu_i \mathbb{1}_{\{a_{t,i} < \theta_c\}} = \sum_{i \in S_t} \mu_i$. This establishes the regret of $\pi$ on $P$ is same as regret of $\pi'$ on $f(P)$ and we get $\mathcal{R}(\mathcal{P}_{MP}) \le \mathcal{R}(\mathcal{P}^c_{CSB})$. Similarly, we can establish the other direction of the proposition and get $\mathcal{R}(\mathcal{P}^c_{CSB}) \le \mathcal{R}(\mathcal{P}_{MP})$. Thus we conclude $\mathcal{R}(\mathcal{P}^c_{CSB}) = \mathcal{R}(\mathcal{P}_{MP})$. $\qquad\square$

### A.4 Proof of Theorem 1

Let $M, \nabla_m$ and $W_\delta$ be defined as in Section 3.2.1. We use the following results to prove the theorem.

**Theorem 3.** *Let $\hat{\theta}_c$ be allocation equivalent to $\theta_c$ for instance $(\boldsymbol{\mu}, \theta_c, Q)$. Then, the expected regret of the regret minimization phase of **CSB-ST** for $T$ rounds is upper bounded as*

$$\mathbb{E}[\mathcal{R}_T] \le O\left((\log T)^{2/3}\right) + \sum_{i \in [K] \setminus [K-M]} \frac{(\mu_i - \mu_{K-M})\log T}{d(\mu_{K-M}, \mu_i)}. \tag{5}$$

*Proof.* As $\hat{\theta}_c$ is the allocation equivalent to $\theta_c$, the instances $(\boldsymbol{\mu}, \theta_c, Q)$ and $(\boldsymbol{\mu}, \hat{\theta}_c, Q)$ have same minimum loss. Also, by the equivalence established is Proposition 1, the regret minimization phase of **CSB-ST** is solving a MP-MAB instance. Then we can directly apply Theorem 1 of Komiyama et al. (2015) to obtain the regret bounds by setting $k = K - M$ and noting that we are in the loss setting and a mistake happens when a arm $i \in [K] \setminus [K-M]$ is in selected superarm. $\qquad\square$

**Theorem 4.** *With probability at least $1 - \delta$, the expected cumulative regret of* **CSB-ST** *is upper bounded as*

$$\mathbb{E}\left[\mathcal{R}_T\right] \leq W_\delta \log_2\left(|\Theta|\right)\nabla_m + O\left((\log T)^{2/3}\right) + \sum_{i \in [K] \setminus [K-M]} \frac{(\mu_i - \mu_{K-M}) \log T}{d(\mu_{K-M}, \mu_i)}.$$

*Proof.* **CSB-ST** has two phases: threshold estimation and loss minimization. Threshold estimation runs for at most $W_\delta \log_2\left(|\Theta|\right)$ rounds and returns an allocation equivalent threshold with probability at least $1 - \delta$. The maximum regret incurred in this phase is $W_\delta \log_2\left(|\Theta|\right)\nabla_m$. Given that the threshold estimated in the threshold estimation phase is correct, the regret incurred in the regret minimization phase is given by Theorem 3. Thus the expected regret of **CSB-ST** is given by the sum of regret incurred in these two phases and holds with probability at least $1 - \delta$. $\qquad\square$

**Theorem 1.** *Let $\mu_1 \geq \epsilon > 0$, $W_\delta = \log(\log_2(|\Theta|)/\delta)/\max\{1, \lfloor Q \rfloor\} \log(1/(1-\epsilon))$, $\mu_{K-M} < \mu_{K-M+1}$, and $T > W_\delta \log_2(|\Theta|)$. Set $\delta = T^{-(\log T)^{-\alpha}}$ in* **CSB-ST** *such that $\alpha > 0$. Then the regret of* **CSB-ST** *is upper bounded as*

$$\mathbb{E}\left[\mathcal{R}_T\right] \leq W_\delta \log_2\left(|\Theta|\right)\nabla_m + O\left((\log T)^{2/3}\right) + \sum_{i \in [K] \setminus [K-M]} \frac{(\mu_i - \mu_{K-M}) \log T}{d(\mu_{K-M}, \mu_i)}.$$

*Proof.* The bound follows from Theorem 4 by setting $\delta = 1/T$ and unconditioning the expected regret obtained in the regret minimization phase of **CSB-ST** . $\qquad\square$

# B  Proofs related to Section 4

## B.1   Proof of Proposition 2

**Proposition 2.** *Let $P = (\boldsymbol{\mu}, \boldsymbol{\theta}, Q) \in \mathcal{P}_{CSB}$. Then the optimal allocation for $P$ is a solution of $KP(\boldsymbol{\mu}, \boldsymbol{\theta}, Q)$ problem.*

*Proof.* Assigning $\theta_i$ fraction of resources to the arm $i$ reduces the total mean loss by amount $\mu_i$. Our goal is to allocate resources such that total mean loss is minimized i.e., $\min_{\boldsymbol{a} \in \mathcal{A}_Q} \sum_{i \in [K]} \mu_i \mathbb{1}_{\{a_i < \theta_i\}}$. Note that the maximization version of same optimization problem is $\max_{\boldsymbol{a} \in \mathcal{A}_Q} \sum_{i \in [K]} \mu_i \mathbb{1}_{\{a_i \geq \theta_i\}}$ which is same as solving a 0-1 knapsack with capacity $Q$ where item $i$ has value $\mu_i$ and weight $\theta_i$. $\qquad\square$

## B.2   Proof of Lemma 3

**Lemma 3.** *Let $\gamma > 0$ and $\forall i \in [K] : \hat{\theta}_i \in [\theta_i, \theta_i + \gamma]$. Then for any $\boldsymbol{\mu} \in [0,1]^K$ and $Q$, the $\boldsymbol{\theta}$ and $\hat{\boldsymbol{\theta}}$ are allocation equivalent.*

*Proof.* Let $L^\star = \{i : a_i^\star < \theta_i\}$ and $r = Q - \sum_{i:a_i^\star \geq \theta_i} \theta_i$. If resource $r$ is allocated to any arm $i \in L^\star$, minimum value of mean loss will not change as $r < \min_{i \in L^\star} \theta_i$. If we can allocate $\gamma = r/K$ fraction of $r$ to each arm $i \in K$, the minimum mean loss still remains same. If estimated threshold of every arm $i \in K$ lies in $[\theta_i, \theta_i + \gamma]$ then using Theorem 3.2 of Hifi and Mhalla (2013), $KP(\boldsymbol{\mu}, \boldsymbol{\theta}, Q)$ and $KP(\boldsymbol{\mu}, \hat{\boldsymbol{\theta}}, Q)$ has the same optimal solution because of having the same mean loss for both the problem instances. $\qquad\square$

## B.3   Proof of Lemma 4

**Lemma 4.** *Let $(\boldsymbol{\mu}, \boldsymbol{\theta}, Q)$ be an instance of CSB such that $\gamma > 0$ and $\mu_1 \geq \epsilon > 0$. Then with probability at least $1 - \delta$, the number of rounds needed by threshold estimation phase of* **CSB-DT** *to find the allocation equivalent for threshold vector $\boldsymbol{\theta}$ is bounded as*

$$T_{\theta_d} \leq \frac{K \log(K \log_2(\lceil 1 + 1/\gamma \rceil)/\delta)}{\max\{1, \lfloor Q \rfloor\} \log(1/(1-\epsilon))} \log_2(\lceil 1 + 1/\gamma \rceil).$$

*Proof.* For any arm $i \in [K]$, we want $\hat{\theta}_i \in [\theta_i, \theta_i + \gamma]$. As $\theta_i \in (0, 1]$, we can divide interval $[0, 1]$ into a discrete set $\Theta \doteq \{0, \gamma, 2\gamma, \ldots, 1\}$ and note that $|\Theta| = \lceil 1 + 1/\gamma \rceil$. As search space is reduced by half in each change of $\hat{\theta}_i$, the maximum change in $\hat{\theta}_i$ is upper bounded by $\log_2 |\Theta|$ to make sure that $\hat{\theta}_i \in [\theta_i, \theta_i + \gamma]$. When $\hat{\theta}_i$ is underestimated and no loss is observed for consecutive $W_\delta$ rounds, a mistake happens by assuming that current allocation is overestimated. We set $W_\delta$ such that the probability of estimating wrong $\hat{\theta}_i$ is small and bounded as follows:

$$\mathbb{P}\left\{\text{No loss is observed for consecutive } W_\delta \text{ rounds when } \hat{\theta}_i \text{ is underestimated}\right\}$$

$$= (1 - \mu_i)^{W_\delta} \quad \left(\text{as } (1 - \mu_i) \text{ is the probability of not observing loss at arm } i\right)$$

$$\leq (1 - \epsilon)^{W_\delta} \quad \left(\text{since } \forall i \in [K] : \mu_i > \epsilon\right)$$

Since we are doing binary search, the algorithm goes through at most $\log_2(|\Theta|)$ underestimates of $\theta_i$. Let $I$ denote the set of indices of these underestimates in $\Theta$

$$\mathbb{P}\left\{\text{No loss is observed for consecutive } W_\delta \text{ rounds when } \hat{\theta}_i \text{ is underestimated}\right\}$$

$$\leq \sum_{i \in I} \mathbb{P}\left\{\text{No loss is observed for consecutive } W_\delta \text{ rounds when } \hat{\theta}_i \text{ is underestimated}\right\}$$

$$\leq (1 - \epsilon)^{W_\delta} \log_2(|\Theta|)$$

Next, we will bound the probability of making mistake for any of the arm. That is given by

$$\mathbb{P}\left\{\exists i \in [K], \hat{\theta}_i \in \Theta : \text{No loss is observed for consecutive } W_\delta \text{ rounds when } \hat{\theta}_i \text{ is underestimated}\right\}$$

$$\leq \sum_{i=1}^{K} \mathbb{P}\left\{\exists \hat{\theta}_i \in \Theta : \text{No loss is observed for consecutive } W_\delta \text{ rounds when } \hat{\theta}_i \text{ is underestimated}\right\}$$

$$\leq K(1 - \epsilon)^{W_\delta} \log_2(|\Theta|)$$

As we are interested in bounding the above probability of making a mistake by $\delta$ for all arms, we have the following expression,

$$K(1 - \epsilon)^{W_\delta} \log_2(|\Theta|) \leq \delta \implies (1 - \epsilon)^{W_\delta} \leq \delta/K \log_2(|\Theta|)$$

Taking log both side, we get

$$W_\delta \log(1 - \epsilon) \leq \log(\delta/K \log_2(|\Theta|)) \implies W_\delta \log\left(1/(1 - \epsilon)\right) \geq \log(K \log_2(|\Theta|)/\delta)$$

$$\implies W_\delta \geq \frac{\log(K \log_2(|\Theta|)/\delta)}{\log\left(1/(1 - \epsilon)\right)}$$

We set

$$W_\delta = \frac{\log(K \log_2(|\Theta|)/\delta)}{\log\left(1/(1 - \epsilon)\right)} \tag{6}$$

Therefore, the minimum number of rounds needed for an arm $i$ to correctly find $\hat{\theta}_i$ with probability at least $1 - \delta$ is upper bounded by $W_\delta \log_2(|\Theta|)$. **CSB-DT** can simultaneously estimate threshold for at least $\max\{1, \lfloor Q \rfloor\}$ arms by exploiting the fact that $\hat{\theta}_i \leq 1$. The threshold for $K$ arms needs to be estimate, hence, minimum rounds needed to correctly find all $\hat{\theta}_i \in [\theta_i, \theta_i + \gamma]$ with probability at least $1 - \delta$ is $KW_\delta \log_2(|\Theta|)/\max\{1, \lfloor Q \rfloor\}$. $\qquad \square$

### B.4 Proof of Proposition 3

**Equivalence of CSB with different thresholds and Combinatorial Semi-Bandit**

In stochastic Combinatorial Semi-Bandits (CoSB), a learner can play a subset from $K$ arms in each round, also known as superarm, and observes the loss from each arm played (Chen et al., 2013, 2016; Wang and Chen, 2018). The size of superarm can vary, and the mean loss of a superarm only depends on the mean of its constituent arms. The goal of the learner is to select a superarm that has the smallest loss. A policy in CoSB selects a superarm in each round based on the past information. The

performance of a policy is measured in terms of regret defined as the difference between cumulative loss incurred by the policy and that incurred by playing an optimal superarm in each round. Let $(\boldsymbol{\mu}, \mathcal{I}) \in [0,1]^K \times 2^{[K]}$ denote an instance of CoSB where $\boldsymbol{\mu}$ denote the mean loss vector, and $\mathcal{I}$ denotes the set of superarms. Let $\mathcal{P}_{\text{CSB}}^d \subset \mathcal{P}_{\text{CSB}}$ denote the set of CSB instances with a different threshold for arms. For any $(\boldsymbol{\mu}, \boldsymbol{\theta}, Q) \in \mathcal{P}_{\text{CSB}}^d$ with $K$ arms and known threshold $\boldsymbol{\theta}$, let $(\boldsymbol{\mu}, \mathcal{I})$ be an instance of CoSB with $K$ arms and each arm has the same Bernoulli distribution as the corresponding arm in the CSB instance. Let $\mathcal{P}_{\text{CoSB}}$ denote set of resulting CoSB problems and $g : \mathcal{P}_{\text{CSB}} \to \mathcal{P}_{\text{CoSB}}$ denote the above transformation.

Let $\pi$ be a policy on $\mathcal{P}_{\text{CoSB}}$. $\pi$ can also be applied on any $(\boldsymbol{\mu}, \boldsymbol{\theta}, Q) \in \mathcal{P}_{\text{CSB}}^d$ with known $\boldsymbol{\theta}$ to decide which set of arms are allocated resource as follows: in round $t$, let information $(L_1, Y_1, L_2, Y_2, \ldots, L_{t-1}, Y_{t-1})$ collected on an CSB instance, where $L_s$ is the set of arms where no resource is applied and $Y_s$ is the samples observed from these arms, is given to $\pi$ which returns a set $L_t$. Then all arms other than arms in $L_t$ are given resource equal to their estimate threshold. Let this policy on $(\boldsymbol{\mu}, \boldsymbol{\theta}, Q) \in \mathcal{P}_{\text{CSB}}^d$ is denoted as $\pi'$. In a similar way a policy $\beta$ on $\mathcal{P}_{\text{CSB}}$ can be adopted to yield a policy for $\mathcal{P}_{\text{CoSB}}$ as follows: in round $t$, the information $(S_1, Y_1, S_2, Y_2, \ldots, S_{t-1}, S_{t-1})$, where $S_s$ is the superarm played in round $s$ and $Y_s$ is the associated loss observed from each arms in $S_s$, collected on an CoSB instance is given to $\pi$. Then $\pi$ returns a set $S_t$ where no resources has allocated. The superarm corresponding to $S_t$ is then played. Let this policy on $\mathcal{P}_{\text{CoSB}}$ be denoted by $\beta'$. Note that when $\boldsymbol{\theta}$ is known, the mapping is invertible.

**Proposition 3.** *The CSB problem with known threshold vector $\boldsymbol{\theta}$ is regret equivalent to a Combinatorial Semi-Bandits where Oracle uses $KP(\boldsymbol{\mu}, \boldsymbol{\theta}, Q)$ to identify the optimal subset of arms.*

*Proof.* Let $\pi$ be a policy on $P := (\boldsymbol{\mu}, \boldsymbol{\theta}, Q) \in \mathcal{P}_{\text{CSB}}^d$. The regret of policy $\pi$ on $P$ is given by

$$\mathcal{R}_T(\pi, P) = \sum_{t=1}^{T} \left( \sum_{i=1}^{K} \mu_i \mathbb{1}_{\{a_{t,i} < \theta_i\}} - \sum_{i=1}^{K} \mu_i \mathbb{1}_{\{a_i^\star < \theta_i\}} \right),$$

where $\boldsymbol{a}^\star$ is the optimal allocation for $P$. Consider $g(P) = (\boldsymbol{\mu}, \mathcal{I}) \in \mathcal{P}_{\text{CoSB}}$ where $\boldsymbol{\mu}$ is the same as in $P$ and $\mathcal{I}$ contains all superarms (set of arms) for which resource allocation is feasible. The regret of policy $\pi'$ on $g(P)$ is given by

$$\mathcal{R}_T(\pi', g(P)) = \sum_{t=1}^{T} \left( l(S_t, \boldsymbol{\mu}) - l(S^\star, \boldsymbol{\mu}) \right)$$

where $S_t$ is the superarm played in round $t$, $S^\star$ is optimal superarm, and $l$ returns mean loss. Note that outcomes of $l(S, \boldsymbol{\mu})$ only depends on mean loss of constituents arms of the superarm $S$. In our setting, $l(S, \boldsymbol{\mu}) = \sum_{i \in S} \mu_i$ where $S = \{i : a_i < \theta_i\}$ for allocation $\boldsymbol{a} \in \mathcal{A}_Q$. It is clear that $\sum_{i=1}^{K} \mu_i \mathbb{1}_{\{a_i^\star < \theta_i\}} = l(S^\star, \boldsymbol{\mu})$. Let $L_t$ be the set of arms where no resource is allocated by $\pi$ in round $t$. Since, loss is only incurred for arms in the set $L_t$, we have $\sum_{i=1}^{K} \mu_i \mathbb{1}_{\{a_{t,i} < \theta_c\}} = \sum_{i \in L_t} \mu_i$. From the definition of $\pi'$, notice that in round $t$, $\pi'$ selects superarm $S_t = L_t$, i.e., set of arms returned by $\pi$ for which no resourced are applied. Hence $\sum_{i=1}^{K} \mu_i \mathbb{1}_{\{a_{t,i} < \theta_c\}} = \sum_{i \in S_t} \mu_i$. This establishes the regret of $\pi$ on $P$ is same as regret of $\pi'$ on $g(P)$ and we get $\mathcal{R}(\mathcal{P}_{\text{CoSB}}) \leq \mathcal{R}(\mathcal{P}_{\text{CSB}}^d)$. Similarly, we can establish the other direction of the proposition and get $\mathcal{R}(\mathcal{P}_{\text{CSB}}^d) \leq \mathcal{R}(\mathcal{P}_{\text{CoSB}})$. Thus we conclude $\mathcal{R}(\mathcal{P}_{\text{CSB}}^d) = \mathcal{R}(\mathcal{P}_{\text{CoSB}})$. $\qquad \square$

## B.5  Proof of Theorem 2

Let $\nabla_{\boldsymbol{a}}$ and $\nabla_m$ be defined as in Section 3.2.1. Let $S_{\boldsymbol{a}} = \{i : a_i < \theta_i\}$ for a feasible allocation $\boldsymbol{a}$, $k^\star = |S_{\boldsymbol{a}^\star}|$ and $W_\delta$ be same as in Section 4.1.1. Note that we will never be able to sample a $\hat{\mu}_i(t)$ to be precisely the true value $\mu_i$ using Beta distribution. We need to consider the $\eta$-neighborhood of $\mu_i$, and such $\eta$ term is common in the analysis of most Thompson Sampling based algorithms (see Wang and Chen (2018) for more details). We need the following results to prove the theorem.

**Theorem 5.** *Let $\hat{\boldsymbol{\theta}}$ be allocation equivalent to $\boldsymbol{\theta}$ for instance $(\boldsymbol{\mu}, \boldsymbol{\theta}, Q)$. Then, the expected regret of the regret minimization phase of* **CSB-DT** *in $T$ rounds is upper bounded as* $\left( \sum_{i \in [K]} \max_{S_{\boldsymbol{a}} : i \in S_{\boldsymbol{a}}} \frac{8|S_{\boldsymbol{a}}| \log T}{\nabla_{\boldsymbol{a}} - 2(k^{\star 2} + 2)\eta} \right) + \left( \frac{K(K-N)^2}{\eta^2} + \frac{8\alpha_1}{\eta^2} \left( \frac{4}{\eta^2} + 1 \right)^{k^\star} \log \frac{k^\star}{\eta^2} + 3K \right) \nabla_m$ *for any $\eta$ such that $\forall \boldsymbol{a} \in \mathcal{A}_Q, \nabla_{\boldsymbol{a}} > 2(k^{\star 2} + 2)\eta$ and $\alpha_1$ is a problem independent constant.*

*Proof.* Once the allocation equivalent to $\boldsymbol{\theta}$ is known, the regret minimization problem is equivalent to solving a Combinatorial Semi-Bandit problem (from Proposition 3). The proof follows by verifying Assumptions $1 - 3$ in Wang and Chen (2018) for the Combinatorial Semi-Bandit problem and applying their regret bounds. Assumption 1 states that the mean of a superarm depends only on the means of its constituents arms (Assumption 1) and distributions of the arms are independent (Assumptions 3). It is clear that both of these assumptions hold for our case. We next proceed to verify Assumption 2. For fix allocation $\boldsymbol{a} \in \mathcal{A}_Q$, the mean loss incurred from loss vector $\boldsymbol{\mu}$ is given by $l(S, \boldsymbol{\mu}) = \sum_{i \in S} \boldsymbol{\mu}_i$ where $S = \left\{ i : a_i < \hat{\theta}_i \right\}$. For any loss vectors $\boldsymbol{\mu}$ and $\boldsymbol{\mu}'$, we have

$$
\begin{aligned}
l(S, \boldsymbol{\mu}) - l(S, \boldsymbol{\mu}') &= \sum_{i \in S} (\mu_i - \mu_i') \\
&= \sum_{i=1}^{K} \mathbb{1}_{\left\{ a_i < \hat{\theta}_i \right\}} (\mu_i - \mu_i') \qquad \left( \text{as} \sum_{i \in S} \mu_i = \sum_{i=1}^{K} \mu_i \mathbb{1}_{\left\{ a_i < \hat{\theta}_i \right\}} \right) \\
&\leq \sum_{i=1}^{K} (\mu_i - \mu_i') \leq \sum_{i=1}^{K} |\mu_i - \mu_i'| = B \parallel \boldsymbol{\mu} - \boldsymbol{\mu}' \parallel_1
\end{aligned}
$$

where $B = 1$. Also, note that in the regret minimization phase, the allocation to each arm remains the same in each round ($\hat{\theta}_i$ is given to each arm $i \in [K] \setminus L_t$). Thus we are solving a Combinatorial Semi-Bandit with parameter $B = 1$ in the regret minimization phase. By applying Theorem 1 in Wang and Chen (2018), we get the desired bounds. □

**Theorem 6.** *With probability at least $1 - \delta$, the expected cumulative regret of* **CSB-DT** *is upper bounded as* $\mathbb{E}[\mathcal{R}_T] \leq \left( \frac{K W_\delta \log_2 (\lceil 1+1/\gamma \rceil)}{\max\{1, Q\}} \right) \nabla_m + \left( \sum_{i \in [K]} \max_{S_{\boldsymbol{a}} : i \in S_{\boldsymbol{a}}} \frac{8 |S_{\boldsymbol{a}}| \log T}{\nabla_{\boldsymbol{a}} - 2(k^{\star 2} + 2)\eta} \right) + \left( \frac{K(K-N)^2}{\eta^2} + \frac{8\alpha_1}{\eta^2} \left( \frac{4}{\eta^2} + 1 \right)^{k^\star} \log \frac{k^\star}{\eta^2} + 3K \right) \nabla_m.$

*Proof.* The threshold estimation phase runs for at most $K W_\delta \log_2 (\lceil 1 + 1/\gamma \rceil)/ \max\{1, Q\}$ rounds and finds an allocation equivalent threshold with probability $1 - \delta$. The regret incurred by this phase is $(K W_\delta \log_2 (\lceil 1 + 1/\gamma \rceil)/ \max\{1, Q\}) \nabla_m$ which form the first part of the bounds. Once an allocation equivalent threshold is found, the upper bound on expected regret incurred in the regret minimization phase is given by Theorem 5. Thus the regret of **CSB-DT** is given by sum of these two quantities holds with probability at least $(1 - \delta)$. □

**Theorem 2.** *Let $(\boldsymbol{\mu}, \boldsymbol{\theta}, Q) \in \mathcal{P}_{CSB}$ such that $\gamma > 0$, $\mu_1 \geq \epsilon$, and $T > W_\delta \log_2(\lceil 1 + 1/\gamma \rceil)$. Set $\delta = T^{-(\log T)^{-\alpha}}$ in* **CSB-DT** *such that $\alpha > 0$. Then the expected regret of* **CSB-DT** *is upper bounded as*

$$
\begin{aligned}
\mathbb{E}[\mathcal{R}_T] \leq & \left( \frac{K W_\delta \log_2 (\lceil 1+1/\gamma \rceil)}{\max\{1, \lfloor Q \rfloor\}} \right) \nabla_m + \left( \sum_{i \in [K]} \max_{S_{\boldsymbol{a}} : i \in S_{\boldsymbol{a}}} \frac{8 |S_{\boldsymbol{a}}| \log T}{\nabla_{\boldsymbol{a}} - 2(k^{\star 2} + 2)\eta} \right) + \\
& \left( \frac{K(K-Q)^2}{\eta^2} + \frac{8\alpha_1}{\eta^2} \left( \frac{4}{\eta^2} + 1 \right)^{k^\star} \log \frac{k^\star}{\eta^2} + 3K \right) \nabla_m,
\end{aligned}
$$

*for any $\eta$ such that $\forall \boldsymbol{a} \in \mathcal{A}_Q, \nabla_{\boldsymbol{a}} > 2(k^{\star 2} + 2)\eta$ and a problem independent constant $\alpha_1$.*

*Proof.* The bound follows from Theorem 6 by setting $\delta = 1/T$ and unconditioning the expected regret obtained in the regret minimization phase of **CSB-DT**. □

## C  Additional details from Section 5

### C.1  Algorithm: **CSB-DT-UCB**

In our experiments, we used an UCB index-based algorithm named **CSB-DT-UCB** for comparing cumulative regret with Thompson Sampling based algorithm **CSB-DT**. **CSB-DT-UCB** works as follows: it keeps track of number of losses ($S_i$) and number of observations ($N_i$) for each arm $i \in [K]$. At round $t$, it computes lower bound of mean losses of all arms (Line 23). We use the lower bound due to our loss setting. The remaining part of **CSB-DT-UCB** is same as **CSB-DT**.

---

**CSB-DT-UCB** UCB based Algorithm for solving the CSB problem with Different Threshold

---

**Input:** $K, Q, \delta, \epsilon, \gamma$

\\ **Threshold Estimation Phase** \\

1: Initialize $\forall i \in [K] : \theta_{l,i} = 0, \theta_{u,i} = 1, \theta_{g,i} = 0, C_i = 0$.
2: Set $T_{\theta_d} = 0, W_\delta = \log(K \log_2(\lceil 1 + 1/\gamma \rceil)/\delta)/\log(1/(1 - \epsilon))$
3: $\forall i \in [\lfloor Q \rfloor] :$ allocate $\hat{\theta}_i = 0.5$ resource. $\forall j \in [\lfloor Q \rfloor + 1, K] :$ allocate $\hat{\theta}_j = \frac{Q - \lfloor Q \rfloor/2}{K - \lfloor Q \rfloor}$ resources
4: **while** $\theta_{g,i} = 0$ for any $i \in [K]$ **do**
5:     **for** $i = 1, \ldots, K$ **do**
6:         **if** loss observe for arm $i$ with $\theta_{g,i} = 0$ and $\theta_{l,i} < \hat{\theta}_i$ **then**
7:             Set $\theta_{l,i} = \hat{\theta}_i, \hat{\theta}_i = (\theta_{u,i} + \theta_{l,i})/2, C_i = 0$. If available allocate resource $\hat{\theta}_i$
8:         **else**
9:             If allocated resources is $\hat{\theta}_i$ then reset $C_i = C_i + 1$
10:           **if** $C_i = W_\delta$ and $\theta_{g,i} = 0$ **then**
11:              Set $\theta_{u,i} = \hat{\theta}_i, \hat{\theta}_i = (\theta_{u,i} + \theta_{l,i})/2, C_i = 0$. If available allocate resource $\hat{\theta}_i$
12:              If $\theta_{u,i} - \theta_{l,i} \leq \gamma$ then set $\theta_{g,i} = 1$ and $\hat{\theta}_i = \theta_{u,i}$
13:           **end if**
14:         **end if**
15:     **end for**
16:     **while** free resources are available **do**
17:         Allocate $\hat{\theta}_i$ resources to a new randomly chosen arm $i$ from the arms having $\theta_{g,i} = 1$
18:     **end while**
19:     $T_{\theta_d} = T_{\theta_d} + 1$
20: **end while**

\\ **Regret Minimization Phase** \\

21: $\forall i \in [K] : S_i = 0, N_i = 1$
22: **for** $t = T_{\theta_d} + 1, T_{\theta_d} + 2, \ldots, T$ **do**
23:     $\forall i \in [K] : \hat{\mu}_i(t) \leftarrow \max\left\{ \frac{S_i}{N_i} - \sqrt{\frac{1.5 \log(t)}{N_i}}, 0 \right\}$
24:     $L_t \leftarrow \text{Oracle}\big(KP(\hat{\boldsymbol{\mu}}(t), \hat{\boldsymbol{\theta}}, Q)\big)$
25:     $\forall i \in L_t :$ allocate no resource to arm $i$. $\forall j \in K \setminus L_t :$ allocate $\hat{\theta}_j$ resources to arm $j$
26:     $\forall i \in L_t :$ observe $X_{t,i}$. Update $S_i = S_i + X_{t,i}$ and $N_i = N_i + 1$
27: **end for**

---

### C.2 Computation complexity of 0-1 Knapsack when items have fractional weight and value

Even though $KP(\boldsymbol{\mu}, \boldsymbol{\theta}, Q)$ is NP-Hard problem; it can be solved by a pseudo-polynomial time algorithm[2] using dynamic programming with the time complexity of O($KQ$). But such an algorithm for $KP(\boldsymbol{\mu}, \boldsymbol{\theta}, Q)$, works when the value and weight of items are integers. In case of $\mu_i$ and $\theta_i$ are fractions, they need to convert these to integers with the desired accuracy by multiplying by large value $S$. The time complexity of solving $KP(S\boldsymbol{\mu}, S\boldsymbol{\theta}, SQ)$ is O($KSQ$) as a new capacity of Knapsack is $SQ$. Therefore, the time complexity of solving $KP(S\boldsymbol{\mu}, S\boldsymbol{\theta}, SQ)$ in each of the $T$ rounds is O($TKSQ$).

Note that the empirical mean losses do not change drastically in consecutive rounds in practice (except initial rounds). As solving 0-1 knapsack is computationally expensive, we can solve it after $N$ rounds. We use $S = 10^4$ and $N = 20$ in our experiments involving different thresholds.