[Reviews · NeurIPS 2019]

Reviewer 1



The authors consider a novel bandit setting where the rewards are censored which arise in a wide-range of applications. Classically, censored data is ubiquitous in medical settings when estimating effectiveness of treatments when patients may drop out of the study. In particular resources are allocated to the arms and depending on arm-dependent thresholds we may or may not observe the loss of the arm. The goal is to both estimate the arm dependent thresholds and the optimal allocation of the resources to the arms which minimizes the expected regret. Comments: Overall, the paper is well-written and is easy to follow. Splitting it into same thresholds and different thresholds and showing the corresponding reductions to the multi-play and its generalized version of combinatorial semi-bandit helps with the development. Also, algorithms for both the settings are shown in the paper dubbed CSB-ST and CSB-DT for same and different thresholds correspondingly. Various connections are shown to related problems such as bandits with knapsacks and contrasts are clearly drawn. Censored settings are of vital interest in practical problems and have not yet been thoroughly studied and this work should help future work in this direction.

Reviewer 2



--------- post rebuttal comments: After carefully reading the comments from the other reviewers and the responses provided by the authors on all the comments, I will keep my score as 5, as I still consider that this a very modest contribution to the literature. That said, if the paper is accepted, I believe that the authors should be able to polish it so as to make it grammatically and mathematically free of typos and ambiguity. --------- a) summary of the content: In this paper, the authors introduce a variant of a semi-bandits problem in order to model some resource allocation problems with censored feedback. The learner is given a fixed amount of a divisible resource which can be allocated to any of $K$ arms at the beginning of each round (the amount of resource is being replenished between each round). The loss observed from a given arm $i$ is random and depends on the amount of resource allocated to it: It is zero if the amount exceeds a constant unknown threshold $\theta_i \in (0,1]$, and is the outcome of a Bernoulli random variable with unknown constant parameter $\mu_i \in [0,1]$ otherwise. The goal of the learner is to find a feasible resource allocation strategy at every round so as to minimize the expected cumulative loss over $T$ rounds, and where the performance measure of a policy is the expected (pseudo) regret over the priod $T$. In the case the unknown thresholds are identical across arms, the authors show that this problem is equivalent to a multiple-play multi-armed bandits problem, and show otherwise that the problem is equivlent to a combinatorial semi-bandits problem. They use the connections to propose optimal algorithms for the problem and conduct some experimental testing of their performances. b) strengths and weaknesses of the submission. * originality: Sequential resource allocation problems with censored feedback is an important class of problems and this paper looks at a stylized version and show its connection to a multiple-play multi-armed bandits problem in the most simple setting, and to a combinatorial semi-bandits problem otherwise. This is an interesting contribution. * quality: The paper seems to be technically sound. I have gone through most proofs in detail, and although some would benefit with added clarity (see some examples below), I haven't found any main flaws. * clarity: The paper would benefit greatly with a rigorous and systematic editing so as to improve its readability. See below some examples. * significance: In terms of significance, I believe that the main contribution of this work is in establishing the connections to well-known bandits problems. Proofs and techniques are otherwise quite standard. The work may be useful for researchers interested in modeling such sequential resource allocations problems in applications linked to security and defense. * minor details/comments: - p.1, line 9: their => its \- p.1, line 34: the use of the term locations here (for arms) should be clarified as early as possible \- p.1, line 35: armed played => armed is played \- p.2, line 39: their => its \- p.2, line 67-68: the experimental evaluation $\Rightarrow$ experimental evaluations \- p.2, line 71: since we are dealing with divisible resources, number => amount. I would also replace $N$ with another notation such as $Q$ so as to make it clear that this quantity is not necessarily an integer \- p.2, line 73: you mean smaller, right? \- p.2, line 81: may or not => may not be \- p.2, line 84: technically shouldn't you include $K$ as identifying an instance of CSB? \- p.2, line 86: please briefly indicate why you don't consider cases when some arms have identical rates, i.e., $\mu_i=\mu_j$, $i \neq j$ \- p.3, Algorithm 1, line 1: the expectation operator is missing \- p.3, Definition 1: a bit ambiguous as written. I would say: For a given loss vector $\mu$ and resource $N$, we say that thresholds ... \- p.3, Definition 1: is the equality between the argmin a set equality? \- p.3, Lemma 1: do you assume that $N \geq 1$ (note that otherwise $M$ could be 0)?. Assuming $M \geq 1$, $\hat{\theta}_c$ should be defined as $\min \{N/M,1\}$, as $N/M$ can exceed 1 \- p. 3, line 106: the optimal $\Rightarrow$ an optimal (this is just one optimal allocation, there may be many others) \- p.3, line 115: leaner => learner \- p.4, line 119: a playing => playing \- p.4, line 125: denote set => denote the \- p.4, line 132: is denoted => be denoted \- p.4, line 134, last term of the information set: $S_{t-1}$ => $Y_{t-1}$ \- p.4, line 152: shown achieve => shown to achieve \- p.5, Algorithm 2: I understand the saving space of putting the algorithm in such a format but it doesnt help readability. Same for Algorithm 3 - p.5, line 180: the assumption that $\mu_1 \geq \epsilon > 0$ is obviously restrictive. It would have been nice to add it on p.2 when discussions and restrictions about the $\mu's$ were introduced \- p.6, line 230: the optimal => an optimal \- p.6, line 236: optimal => an optimal \- p.6, line 254: give variables ?

Reviewer 3



I found the task is new and interesting. I skimmed the proof and it seems sound to me. The paper is clearly written and would be an important building block for handling censored feedback. Several minor errors: - Line 73 "if the allocated resource is larger...": I suspect it should be "smaller" rather than "larger". - Equation (2): This is asymptotic lower bound. It should be explicitly noted that the inequality holds when T goes infinite - Corollary 3: Since we set delta = 1/T, we have W=O(log T). Hence, we cannot say it is asymptotically optimal simply because MP-TS is. ===After Rebuttal==== Thank you for the rebuttal. I have read the response but it did not change my score. One thing I would like to note is that "asymptotically optimal" does not just mean the regret is O(log T) but means that the regret **equals** the lower bound given in (2). Therefore, you have to make a careful discussion when you insist CSB-ST is indeed optimal in spite of W=O(log T). That is, you have to prove that W is asymptotically ignorable compared to (2). If not, you should not use the word "optimal".

[Author Response · NeurIPS 2019]

We thank the reviewers for the detailed comments and suggestions. Please find our responses below.

**Response to Reviewer** 1**:**

`Performance on real datasets:`
We are in conversation with groups having private access to wildlife poaching data where patrol scheduling to combat
opportunistic crime is of major practical interest. We hope to include an extensive study on several real-world datasets
as part of a journal version of the current submission.

`Number of thresholds smaller than the number of arms case:`
It is an interesting direction and will serve as a non-trivial extension of the current setting. Of course, one can naively
solve it using the different-thresholds case of this paper. But we believe that a smarter, specialized algorithm whose
performance will smoothly depend on how many and how *separated* the thresholds are can potentially be developed.

**Response to Reviewer** 2**:**

`Significance:`
The problem setup we considered has plenty of applications in domains such as police patrolling, poaching control,
medical diagnosis, advertisement budget allocation, among many others. In this paper, we have proposed a novel
framework for resource allocation (Censored Semi-Bandits (CSB)), which directly addresses such practical use cases.
From the most natural way of formulating this problem, it is not at all apparent apriori that it reduces to Multi-Play or
Combinatorial Semi-Bandits setup. Only a deeper understanding of the problem makes this connection explicit, which
we feel is non-trivial. Furthermore, we believe showing such a reduction will help future work in this area. It would
interest other researchers to look into richer models in resource allocation with censored feedback.

`Including $K$ in identifying an instance of CSB:`
Not necessary. As $K = |\mu|$, $K$ is known (implicitly) from an instance of CSB. We will make it clear in the final version.

`Arms with identical rates:`
We only require $\mu_1 \leq \mu_2 \leq \ldots \leq \mu_{K-M-1} < \mu_{K-M} \leq \ldots \leq \mu_K$, i.e., $\mu_{K-M-1}$ and $\mu_{K-M}$ are distinct, so that the
KL-divergence in Theorem 1 is well defined (as required in [22]). This assumption is equivalent to saying that the set of
arms under optimal allocation is unique. Note that CSB-DT does not need such an assumption. We will update this.

`Definition 1:` Thanks for catching this typo. '$\arg\min$' operator should be replaced by 'min.' We will update this.

`Assumptions on $N$ and $M$ in Lemma 1:`
We agree that a lower bound on $N$ is needed to avoid the case $M = 0$. The assumption $N \geq \theta_c$ will ensure this (weaker
than $N \geq 1$). We will state this. However, it is not necessary to define $\hat{\theta}_c$ as $\min\{N/M, 1\}$ to avoid it exceed 1. $\hat{\theta}_c$ can
be allocation equivalent to $\theta_c$ even if $\hat{\theta}_c > 1$ and it does not disturb our analysis.

`Assumption on $\mu_1 \geq \epsilon > 0$:`
The assumption that $\mu_1 \geq \epsilon > 0$ is obviously restrictive. But it holds naturally when only arms with non zero mean loss
are considered for resource allocation. In such setup, the minimum mean loss is at least $\epsilon$ for some $\epsilon > 0$. We will state
it in the revision. It is still interesting to remove this assumption and will take it as future work.

**Response to Reviewer** 3**:**

`Asymptotic lower bound should be explicitly noted:`
Agreed (Theorem 3.1 of [21]). We will make it explicit.

`Asymptotic optimality of the algorithm:`
The asymptotic optimality indeed holds when we set $\delta = 1/T$. We indeed have $W = O(\log T)$, but that does not
invalidate the optimality as can be checked by substitution.

`Joint estimation of threshold and loss:`
A natural algorithm for joint estimation is as follows: One first starts with a threshold vector, observes losses, and
updates the mean loss for each arm. In the following round, the threshold is updated based on the observations in the
previous rounds. The analysis of such an EM type algorithm for simultaneous estimation seems far more involved, but
still doable. We will take it as an extension of this work.

`The tolerance parameter $\gamma$ as input:`
We needed to know a lower bound on $\gamma$ so that we can estimate $\boldsymbol{\theta}$ within some approximation. If we do not know this,
we can use the joint estimation of threshold and loss as explained in the previous response. It will lead to an estimate of
$\boldsymbol{\theta}$ that will eventually fall within the desired range of approximation provided $\gamma > 0$. However, the analysis of this is
delicate, and we aim to take it as an extension of this work. We believe that the condition $\gamma > 0$ is necessary. Otherwise,
sub-linear regret may not be achievable.

[Meta-Review · NeurIPS 2019]

The reviewers largely liked this paper. They also suggest a number of minor edits that would improve the presentation. Please take these into careful consideration when preparing the final version.